# CITING: LARGE LANGUAGE MODELS CREATE CURRICULUM FOR INSTRUCTION TUNING

## ABSTRACT

The recent advancement of large language models (LLMs) has been achieved through a combo of instruction tuning and human alignment. However, building manually crafted instruction datasets and performing human alignment become the bottleneck for scaling the development of LLMs. In this paper, we exploit the idea of leveraging AI models in lieu of humans as the teacher to train student LLMs. Our method is inspired by how human students refine their writing skills by following the rubrics and learning from the revisions offered by their tutors. Specifically, we employ a teacher LLM to create a curriculum for instruction tuning of the student LLM, namely **C**urriculum **I**nstruction **T**un**ING** (CITING). It encompasses two main steps: (1) the teacher LLM crafts the rubrics for evaluating the answers corresponding to various types of questions, and (2) the student LLM learns to follow the rubrics and perform self-correction from the revision made by the teacher. We further iteratively carry out it to embody the procedure of CITING. We compare CITING to a series of state-of-the-art baselines on four datasets. Our method demonstrates strong improvement in terms of articulate, in-depth, and comprehensive by GPT-4 evaluation. Specifically, it achieves an average winning rate of 79.4% over SFT, 73.4% over RLHF, 78.1% over RRHF, and 76.3% over RAFT, respectively.

## 1 INTRODUCTION

Large Language Model (LLM), equipped with instruction tuning (Wei et al., 2021) and learning from human feedback (LHF) (Ouyang et al., 2022), has demonstrated unparalleled proficiency in understanding and generating human-like text (Alberts et al., 2023). Its capabilities span a myriad of applications, including content creation (Abburi et al., 2023), code generation (Vaithilingam et al., 2022), and answering queries (Li et al., 2023). Technically, we usually need to collect high-quality human-written answers corresponding to input questions for instruction tuning. As such, LHF was usually formulated as a *ranking* task for human evaluators because judging the model output quality is much more efficient than writing a high-quality answer from scratch.

However, either building instruction datasets or aligning LLMs with human feedback renders substantial labor and time costs. Researchers are thus motivated to distill the knowledge of advanced LLMs to facilitate the training of a student LLM, which includes building instruction datasets with LLM-generated answers and learning from AI feedback (LAIF):

- **Instruction tuning with synthetic data**. The generated answers from advanced LLMs, e.g., GPT-4, usually match the quality with human answers (OpenAI, 2023). This finding motivates the idea of fine-tuning student LLMs with the output from a teacher LLM (Ho et al., 2022; Magister et al., 2022). Nonetheless, this approach causes suboptimal performances because the synthetic data usually contain hallucinations produced by LLMs (Ji et al., 2023).

- **Learning from AI feedback**. It was reported that LLMs can outperform human evaluators in many text annotation tasks (Gilardi et al., 2023). This suggests the potential for employing AI feedback to enhance the performance of a student LLM by reinforcement learning, namely RLAIF (Bai et al., 2022; Lee et al., 2023). However, RLAIF-based LLMs are still inferior to their RLHF counterparts because AI feedback is essentially a "lossy transmission" of human preference. In addition, these methods are based on reinforcement learning that is sensitive to hyperparameters and is computationally expensive (Yuan et al., 2023).

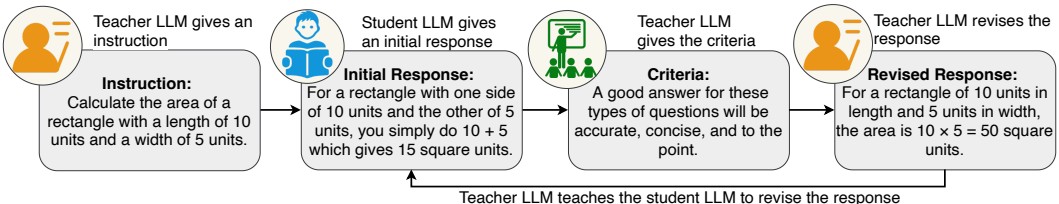

Figure 1: The curriculum instruction tuning process. The teacher LLM teaches the student LLM to revise its response based on criteria and revised responses.

In this paper, we argue that it is crucial to exploit the *generation* capability of the teacher LLM to make LAIF excel. In LHF, human labelers find *ranking* easier than *writing*, but these two tasks actually share a comparable difficulty level for an LLM. With this insight, we propose to rewrite the initial response of the student LLM with a teacher LLM, which further acts as supervision to the student LLM. Our approach draws inspiration from the learning process of humans, mirroring the way students refine their writing skills through practice and mimicking the polished manuscripts from their tutor. To be specific, we employ a

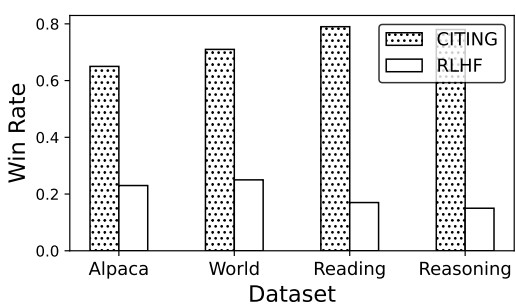

Figure 2: CITING vs. RLHF on four tasks.

teacher LLM to craft a tailored *curriculum* for instruction tuning, which we refer to as **C**urriculum **I**nstruction **T**un**ING** (CITING). The conceptual demonstration is shown in Figure 1. The technical contribution of our method hinges on the following insight:

- **Rubric design with a teacher model**: we employ a teacher LLM to craft the criteria for assessing the quality of student responses corresponding to different types of questions. These criteria also provide supplementary guidance for the student LLM to correct bad answers.

- **Learning to revise**: based on the initial response from the student LLM, we utilize a teacher LLM to provide personalized revisions. Contrasting the revision and the initial answer, the student learns to improve their responses through self-reflection. We can further refine the student by iterating this process.

As shown in Figure 2, we empirically identify that CITING outperforms RLHF by a great margin, achieving an overall winning rate of 73.4% according to GPT-4 evaluation. In the following, we discuss the related papers in Section 2. Then, we elaborate on the details of our method in Section 3. We show more experiment results in Section 4, where CITING showcases remarkable few-shot (Alpaca) and zero-shot (World Knowledge, Reading Comprehension, Commonsense Reasoning) performances compared with baselines.

## 2    RELATED WORK

**LLM Alignment**. It was highlighted that aligning LLMs with human preferences boosts their effectiveness and safety, e.g., reinforcement learning from human feedback (RLHF) (Ouyang et al., 2022). Researchers are then encouraged to mitigate the instability and insufficiency of reinforcement learning (RL) in the human alignment process (Yuan et al., 2023; Dong et al., 2023). On the other hand, the high labor and time cost of LHF also becomes a concern for scaling LLM development. This challenge motivates a series of research in turning large AI models, e.g., GPT-4, to supervise smaller LLMs, termed "Learn from AI Feedback (LAIF)". The prominent examples include reducing the harmfulness of LLMs (Bai et al., 2022) and approximating the performance of RLHF with an AI labeler in lieu of humans (Lee et al., 2023). Nonetheless, LAIF has not yet shown superiority in their performances compared with LHF methods.

**Instruction Tuning**. LLMs can learn to adhere to user requests by learning from handcrafted instruction datasets (Brown et al., 2020; Zhang et al., 2023). An alternative way is to gather instruction data by employing the most capable LLMs to generate answers given many instructions (Wang et al., 2022). Nonetheless, directly applying synthetic instruction datasets introduces defects such as mode collapse due to the hallucinations, low-quality, and imbalanced samples produced by LLMs (Shumailov et al., 2023). Follow-up works aim to refine the synthetic instruction tuning process by prompting the teacher LLM to provide multi-step reasoning rationales (Zelikman et al., 2022) or by progressive training (Hsieh et al., 2023). By contrast, our method leverages the teacher LLM to customize a revision based on the student's answer instead of producing the reference answer from scratch. It mitigates the risk of mode collapse and offers more specific feedback to the student LLM.

# 3 METHOD

## 3.1 SYSTEM OVERVIEW

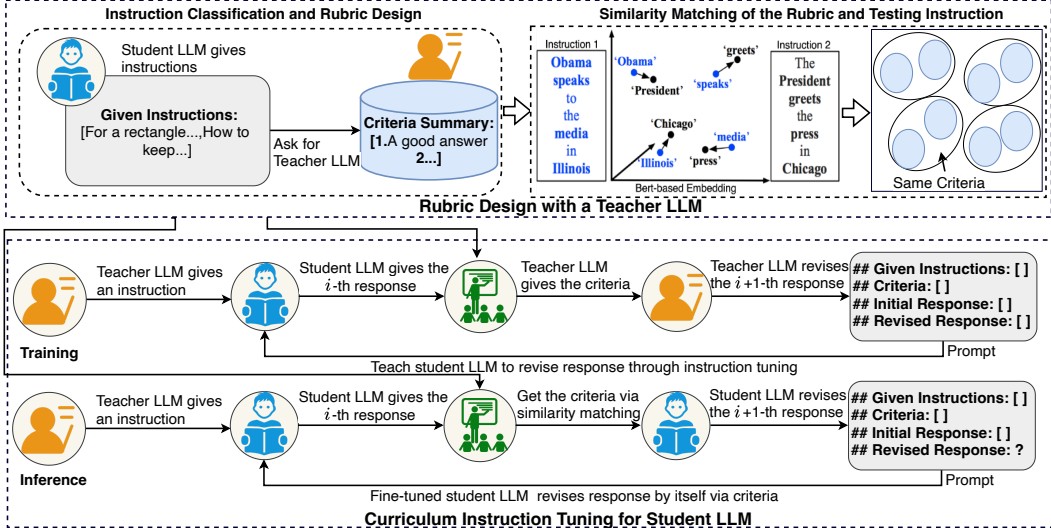

Figure 3: An overview of CITING. It mainly includes two parts: Rubric Design from Teacher LLM and Curriculum Instruction Tuning for Student LLM. In the first part, the teacher LLM designs the rubrics for different types of instruction tasks. In the second part, the teacher LLM teaches the student LLM to revise its initial response based on the rubric.

The instruction tuning task usually inputs an instruction $x$ and outputs a response to imitate the ground truth response $y$. Different from this, our framework includes a criterion $c$ as the input that describes the evaluation standard for the answers. Therefore, we denote our data as $(x, y, c) \sim D$, where $D$ is the distribution of the data.

Based on the input and output of the model introduced above, we propose CITING with a student LLM and a teacher LLM to teach the student LLM to revise its response. The structure overview is shown in Figure 3. The system mainly consists of two parts:

- **Rubric Design by Teacher LLM.** It first employs a teacher LLM to categorize all instructions into multiple types and design the rubric for each type. In the testing phase, a BERT-based similarity matching module is then developed to classify new instructions into these types.

- **Curriculum Instruction Tuning for Student LLM.** The student LLM learns to revise its initial response from teacher LLM based on the criteria from the first part via instruction turning.

## 3.2 RUBRIC DESIGN WITH A TEACHER LLM

To enable the language model to learn to perform self-reflection, it is essential to provide the model with clear criteria for evaluating the quality of its responses to specific instructions. In this section, we will elaborate on how to establish the criteria corresponding to each instruction.

**Instruction classification and rubric design** Employing a language model to give the criteria to each single instruction sample can be cost-prohibitive, posing challenges for scaling it up. To address this concern, we opt to have the language model categorize the instructions within the dataset and generate corresponding criteria for each category. In particular, we sample a subset of instructions from the dataset and create a dedicated prompt to classify them associated with the rubrics.

```
Please classify the following instructions and give good or bad
criteria for each category:
Given instructions: [...]
```

As a result, these instructions are divided into $M$ categories and the criteria of the $i_{th}$ category of instructions is $c_i \sim C$.

**Similarity matching of the rubric and testing instruction** In the testing phase, for a new instruction, we propose to match the rubrics of existing instructions so as to assist the language model in its self-revision process. We leverage sentence-BERT (Reimers & Gurevych, 2019) to encode instructions and rubrics into compact embeddings that are in the same semantic space. Specifically, for each instruction $x_h \sim D_h$ that already has criteria and each new instruction $x_n \sim D_n$, we have

$$e_h = \text{BERT}(x_h), e_n = \text{BERT}(x_n). \tag{1}$$

We hence obtain the embedding set for each category $j$ as $E_j$. Subsequently, we compute the sample-wise similarity for the instruction $x_n$ with $E_j$ and perform mean-pooling to obtain the final similarity score $\text{Score}_j$:

$$\text{Score}_j = \frac{1}{n} \left( \sum_{k=1}^{n} \text{Cosine}(e_n, e_{kh}) \right), \tag{2}$$

where $e_{kh}$ denotes the $k_{th}$ element of $E_j$. Finally, for the instruction $x_n$, we assign the criteria corresponding to the category with the highest similarity score among the M categories.

## 3.3 CURRICULUM INSTRUCTION TUNING FOR STUDENT LLM

By far, we have obtained an instruction tuning dataset $(x, y, c) \sim D$, where $x$ is the instruction, $y$ is the ground truth answer, and $c$ is the criteria for this instruction. The instruction tuning process of `CITING` has two stages, as described below.

**Supervised Fine-tuning** The first stage follows the standard instruction tuning practice, which encourages LLMs to adhere to user requests when returning the outputs. For a provided input instruction, represented as $x$, labelers supply a response representing the desired behavior, consisting of $t$ tokens and denoted as $y = \{y_1, \dots y_t\}$. We train the LLM to yield $f_{\text{SFT}}$ via

$$\mathcal{L}_{SFT} = -\sum_t log P_{f_{\text{SFT}}}(y_t | x, y_1, \dots y_{t-1}). \tag{3}$$

**Curriculum Instruction Tuning with Criteria** We prompt $f_{\text{SFT}}$ to generate initial response $r^{(0)}$ given the instruction $x$. We design a prompt that employs the teacher LLM to improve the current $r^{(0)}$ based on the instruction $x$, the criteria $c$, and the initial responses $r^{(0)}$ to obtain the revised responses $r^{(1)}$:

```
Below is an instruction and its response. In addition, a
criteria for the instruction is given to provide a good or bad
judgment standard for completing this instruction. Please revise
the response according to the given instruction and criteria.
```

---

**Algorithm 1** CITING

---

**Input:** An instruction dataset $x, y \sim D$, which contains instructions $x$ and corresponding ground truth responses $y$; a teacher LLM; number of curriculum instruction tuning $N$; number of rounds of inference $M$.

1: Initialize student LLM $\pi$ with random weights;
   **// Train with SFT**
2: **for** $i$=0,1,2... **do**
3:     Train student LLM with SFT by minimizing equation (3) using instruction dataset;
4: **end for**
   **// Preparation for Criteria**
5: Utilize the teacher LLM to summarize the criteria for each category of input instructions;
6: A BERT-based similarity matching method is employed to obtain criteria for the remaining instructions based on equation (1) and (2).
   **// Curriculum Instruction Tuning**
   **\*\* Training**
7: **for** $k = 0, 1, 2...N$ **do**
8:     Utilize the student LLM model $\pi^{(k)}$ to generate its responses $r^{(k)}$ for instructions $x$ of the $k - th$ iteration;
9:     Employ the teacher LLM to revise the responses $r^{(k)}$ based on $(x, c, r^{(k)})$ and get the revised responses $r^{(k+1)}$;
10:     Fine-tune $\pi^{(k)}$ based on prompt $pr$ and $(x, c, r^{(k)}, r^{(k+1)})$ to obtain $\pi^{(k+1)}$ by minimizing equation (4).
11: **end for**
   **\*\* Inference**
12: **for** $j = 0, 1, 2...M$ **do**
13:     Employ the fine-tuned student LLM $\pi_*$ to generate its responses $r^{(j)}$ for instructions $x$ of the $j - th$ iteration;
14:     Utilize the student LLM $\pi_*$ to revise the responses $r^{(j)}$ based on $(x, c, r^{(j)})$ and get the revised responses $r^{(j+1)}$.
15: **end for**

---

```
Instruction: [ ]
Response: [ ]
Criteria: [ ]
The revised response is:
```

Therefore, we can obtain the instruction tuning dataset $(x, y, c, r^{(0)}, r^{(1)}) \sim D$. The prompt that contains $pr(x, c, r^{(0)})$ is leveraged to supervise the student model $f_{\text{SFT}}$ via instruction tuning, as

$$\mathcal{L}_{\text{CITING}} = -\sum_t log P_{f_{\text{CITING}}^{(0)}}(r_t^{(1)} | pr(x, c, r^{(0)}), r_1^{(1)}, \ldots r_{t-1}^{(1)}), \tag{4}$$

to yield the first-round CITING model $f_{\text{CITING}}^{(0)}$. It is noted that we can iterate this process by (1) prompting the teacher to revise the output from $f_{\text{CITING}}^{(0)}$ to build the curriculum instruction dataset and (2) refining $f_{\text{CITING}}^{(0)}$ by Eq. 4 to obtain the model $f_{\text{CITING}}^{(1)}$. In the experiment section, we will demonstrate how this iterative process facilitates the student.

## 3.4 MODEL TRAINING AND INFERENCE

We summarize the details of the training and inference process of CITING in Algorithm 1. From the algorithm, the student LLM is trained with SFT using the instruction dataset (lines 2-4). Then, we utilize the teacher LLM and a BERT-based similarity matching method to obtain criteria for all instructions (lines 5-6). Further, we train the student LLM $\pi$ by minimizing log-likelihood via curriculum instruction tuning (lines 7-11). Finally, we employ the fine-tuned student LLM $\pi_*$ to revise the response step by step (lines 12-15).

## 4 EXPERIMENTS

### 4.1 EXPERIMENTAL SETTING

**Datasets** We mainly conduct experiments on the following four data sets: **Alpaca** (Taori et al., 2023) is a dataset of 52,000 instructions and demonstrations generated by OpenAI's text-davinci-003 engine. This data set contains various types of questions and responses, such as general knowledge, reading comprehension, logical reasoning, etc. It is widely used to conduct instruction-tuning for language models and make the language model follow instruction better; **World Knowledge** focuses on facts, concepts, entities, relationships, and contextual information about the external environment, history, culture, science, and more. We integrated the NaturalQuestions dataset (Kwiatkowski et al., 2019) and the TriviaQA dataset (Joshi et al., 2017) to form the World Knowledge data set; **Reading Comprehension** is a curated collection of textual passages accompanied by a set of questions. For each question, there exists a corresponding response that can be directly extracted from, or inferred using, the associated passage. In this paper, we mix the SQuAD dataset (Rajpurkar et al., 2018), QuAC dataset (Choi et al., 2018) and BoolQ dataset (Clark et al., 2019) to form the Reading Comprehension dataset; **Commonsense Reasoning** is a curated collection of questions that are designed to assess the ability of machine learning models to perform reasoning based on commonsense knowledge. The questions in this dataset are typically paired with responses, and possibly with explanations or justifications that shed light on the reasoning process. Specifically, our Commonsense Reasoning dataset is a mixture of PIQA dataset (Bisk et al., 2020), OpenBookQA dataset (Mihaylov et al., 2018) and CommonsenseQA dataset (Talmor et al., 2018).

**Evaluation Metrics** Many studies have shown that GPT-4's ability to judge different responses to questions has reached or even exceeded humans (Gilardi et al., 2023). Therefore, in our paper, we employ GPT-4 to mimic human evaluation to compare two random responses and give a comparison between them (win/lose/tie). In order to compare the quality of different responses more comprehensively, we let GPT-4 judge from the following three aspects:

- **Articulate** is to judge how clearly and fluently a response is presented, which looks at the structure, language quality, and overall readability. Specifically, it will judge the quality of the response from the following aspects: (1) Grammar and syntax correctness; (2) Logical flow of information; (3) Avoidance of jargon, or if jargon is used, it is properly explained;

- **In-depth** focuses on how thoroughly a topic or question is addressed. An in-depth response delves into details and nuances, rather than just scratching the surface. Detaily, its evaluation criteria are specifically: (1) Coverage of core principles or concepts; (2) Incorporation of nuanced viewpoints or less-known facts; (3) Demonstrated understanding beyond the basic level;

- **Comprehensive** evaluates the breadth of a response and is about covering a wide range of related facets or sub-topics. Furthermore, it mainly makes specific judgments from the following aspects: (1) Addressing multiple angles or facets of the question; (2) Incorporating various viewpoints or perspectives; (3) Ensuring no major sub-topic or relevant information is left out.

### 4.2 IMPLEMENTATION DETAIL

In our work, LLaMA-7B [1] is employed as the backbone model, which has evolved into a popular research area for LLM studies (Touvron et al., 2023). We fine-tune it with `CITING` algorithm built on Huggingface Library (Muennighoff et al., 2023). Specifically, we divide the Alpaca dataset into a training set, a validation set, and a test set according to the ratio of 8:1:1. We first fine-tune the LLaMA-7B model with the training set and the validation set based on LoRA framework (Hu et al., 2021) using supervised learning. Then, we sample 1,000 instructions and use the fine-tuned model to generate corresponding initial responses. Then, based on GPT-3.5, we revise these initial responses according to the standard of response quality of instruction. We perform multiple fine-tuning with `CITING` algorithm based on these 1000 instructions and the revised response sample pairs.

We evaluate the fine-tuned models on the test set of the Alpaca dataset. Moreover, we evaluate the zero-shot result on the test set of the World Knowledge, Reading Comprehension, and Commonsense Reasoning datasets.

---

[1]https://huggingface.co/decapoda-research/llama-7b-hf

Table 1: Performance comparison over all baselines in four datasets.

| Methods | Alpaca | | | | | | | | | World Knowledge | | | | | | | | |
|---|---|---|---|---|---|---|---|---|---|---|---|---|---|---|---|---|---|---|
| | Articulate | | | In-depth | | | Comprehensive | | | Articulate | | | In-depth | | | Comprehensive | | |
| | Win | Tie | Lose | Win | Tie | Lose | Win | Tie | Lose | Win | Tie | Lose | Win | Tie | Lose | Win | Tie | Lose |
| CITING vs SFT | 75% | 4% | 21% | 70% | 16% | 14% | 66% | 4% | 30% | 78% | 8% | 14% | 80% | 6% | 14% | 65% | 4% | 31% |
| CITING vs RLHF | 65% | 17% | 13% | 58% | 8% | 34% | 74% | 4% | 22% | 74% | 5% | 21% | 75% | 5% | 20% | 63% | 4% | 33% |
| CITING vs RRHF | 73% | 9% | 18% | 66% | 12% | 22% | 65% | 3% | 32% | 82% | 8% | 10% | 84% | 8% | 8% | 64% | 4% | 32% |
| CITING vs RAFT | 69% | 20% | 10% | 61% | 10% | 28% | 76% | 1% | 23% | 80% | 4% | 16% | 78% | 4% | 18% | 66% | 2% | 32% |

| Methods | Reading Comprehension | | | | | | | | | Commonsense Reasoning | | | | | | | | |
|---|---|---|---|---|---|---|---|---|---|---|---|---|---|---|---|---|---|---|
| | Articulate | | | In-depth | | | Comprehensive | | | Articulate | | | In-depth | | | Comprehensive | | |
| | Win | Tie | Lose | Win | Tie | Lose | Win | Tie | Lose | Win | Tie | Lose | Win | Tie | Lose | Win | Tie | Lose |
| CITING vs SFT | 84% | 4% | 12% | 88% | 2% | 10% | 69% | 4% | 27% | 84% | 6% | 10% | 82% | 14% | 4% | 76% | 2% | 22% |
| CITING vs RLHF | 84% | 6% | 10% | 87% | 4% | 9% | 66% | 3% | 31% | 85% | 5% | 10% | 79% | 10% | 11% | 71% | 4% | 25% |
| CITING vs RRHF | 88% | 6% | 6% | 88% | 6% | 6% | 64% | 4% | 32% | 89% | 2% | 9% | 84% | 3% | 13% | 90% | 4% | 6% |
| CITING vs RAFT | 86% | 4% | 10% | 86% | 6% | 8% | 62% | 2% | 36% | 82% | 2% | 16% | 84% | 2% | 14% | 82% | 4% | 14% |

The sequence length, epoch, and learning rate for all the experiments are set to 512, 4, and 1e-5, respectively, while the maximum number of new tokens generated during inference is 512. We use 8 NVIDIA GeForce RTX 3090 GPUs for fine-tuning, training CITING typically costs 4-6 hours.

## 4.3 BASELINES

We compare CITING with zero-shot baselines fine-tuned on LLaMA-7B which share the same backbone with CITING:

**SFT** (Sun et al., 2023) is a fundamental method that straightforwardly fine-tune language models using supervised learning; **RLHF** is successively promoted by (Brown et al., 2020) and (Nakano et al., 2021) to align the core of language models with human preference in reinforcement learning settings. We implement RLHF according to trlx [2]; **RRHF** (Yuan et al., 2023) takes candidate ranking into account, and distinguishes different candidates through pair-wise ranking losses. We implement it with its official code [3]; **RAFT** (Dong et al., 2023) is a new framework introduced to align generative foundation models with human ethics and preferences effectively. Addressing the inefficiencies of previous methods like RLHF, RAFT uses a reward model to select high-quality samples while discarding undesired ones, creating a streaming dataset for model alignment.

## 4.4 MAIN RESULTS

We report the performance of our model and competing baselines in Table 4.3. From this table, we make the following observations:

- CITING consistently achieves the best performance in all four datasets and across all the metrics. Moreover, compared with other baselines, CITING has great advantages on the same dataset distribution (Alpaca) and remarkable superiority on zero-shot tasks (World Knowledge, Reading Comprehension, Commonsense Reasoning). In particular, for the task on World Knowledge, the CITING reaches at least 52%, 55%, and 30% win rate advantages over the baseline in Articulate, In-depth, and Comprehensive metrics, respectively. The results are attributed to the ability of CITING to currically revise its response according to the criteria, which shows remarkable flexibility and superiority.

- Among the four data set tasks, CITING has the most obvious performance advantage on the Common Reasoning dataset and the smallest advantage on the Alpaca dataset. In addition, we can also

---

[2]https://github.com/CarperAI/trlx
[3]https://github.com/GanjinZero/RRHF

Table 2: Performance comparison over different revision round in four datasets.

| Methods | Alpaca | | | | | | | | | World Knowledge | | | | | | | | |
| | Articulate | | | In-depth | | | Comprehensive | | | Articulate | | | In-depth | | | Comprehensive | | |
| | Win | Tie | Lose | Win | Tie | Lose | Win | Tie | Lose | Win | Tie | Lose | Win | Tie | Lose | Win | Tie | Lose |
|---|---|---|---|---|---|---|---|---|---|---|---|---|---|---|---|---|---|---|
| CITING @1 vs SFT | 75% | 4% | 21% | 70% | 16% | 14% | 66% | 4% | 30% | 78% | 8% | 14% | 80% | 6% | 14% | 65% | 4% | 31% |
| CITING @2 vs CITING @1 | 54% | 14% | 33% | 61% | 12% | 27% | 53% | 4% | 43% | 50% | 24% | 26% | 48% | 22% | 30% | 56% | 14% | 40% |
| CITING @3 vs CITING @2 | 50% | 15% | 35% | 57% | 11% | 32% | 55% | 5% | 40% | 47% | 25% | 28% | 50% | 20% | 30% | 54% | 14% | 42% |
| CITING @4 vs CITING @3 | 45% | 17% | 38% | 53% | 11% | 36% | 52% | 6% | 42% | 44% | 22% | 34% | 45% | 21% | 34% | 48% | 15% | 47% |

| Methods | Reading Comprehension | | | | | | | | | Commonsense Reasoning | | | | | | | | |
| | Articulate | | | In-depth | | | Comprehensive | | | Articulate | | | In-depth | | | Comprehensive | | |
| | Win | Tie | Lose | Win | Tie | Lose | Win | Tie | Lose | Win | Tie | Lose | Win | Tie | Lose | Win | Tie | Lose |
|---|---|---|---|---|---|---|---|---|---|---|---|---|---|---|---|---|---|---|
| CITING @1 vs SFT | 84% | 4% | 12% | 88% | 2% | 10% | 69% | 4% | 27% | 84% | 6% | 10% | 82% | 14% | 4% | 76% | 2% | 22% |
| CITING @2 vs CITING @1 | 54% | 10% | 46% | 48% | 16% | 36% | 52% | 4% | 44% | 64% | 1% | 35% | 60% | 14% | 26% | 62% | 16% | 22% |
| CITING @3 vs CITING @2 | 50% | 14% | 46% | 44% | 16% | 40% | 51% | 5% | 44% | 60% | 3% | 37% | 55% | 15% | 30% | 58% | 14% | 28% |
| CITING @4 vs CITING @3 | 44% | 12% | 46% | 43% | 16% | 41% | 45% | 5% | 50% | 54% | 4% | 42% | 52% | 14% | 34% | 52% | 16% | 32% |

observe that CITING performs better than Alpaca on three zero-shot datasets. These observations confirm two aspects: (1) CITING's generalization performance is much stronger than other baselines, which can generate datasets with different distributions; (2) Curriculum instruction tuning plays an important role in enhancing the ability of reasoning for LLM, which leads to remarkable performance in Commonsense Reasoning dataset.

- Across all the three metrics, CITING's performance is most significant on In-depth and worst on Comprehensive. These observations prove two aspects: (1) Based on multiple revisions of curriculum instruction tuning, CITING learns how to think deeply about how to respond to instruction and gains a deeper understanding of the instruction; (2) Compared with other baselines, CITING does not input additional related facets or sub-topic instructions. Therefore, the improvement in Comprehensive is limited.

To conclude, CITING achieves notable performance gains compared with the state-of-the-art baselines in all datasets and across all metrics, which shows the superiority of incorporating the reasoning ability in LLM's revision.

## 4.5 ABLATION STUDY

In this part, we want to explore the impact of different rounds of curriculum instruction tuning on CITING performance, as mentioned in Section 3.3. We report the comparison of adjacent rounds in Table 4.3. From this table, we make the following observations:

- The results show that in multiple rounds of instruction tuning, CITING performance gains are getting smaller and smaller. For example, on the World Knowledge dataset, the winning rate of CITING for instruction tuning dropped from 78% to 44%, and in the end the performance is even close to the previous round of method performance. This shows that although the performance of CITING can be gradually improved through step-by-step curriculum instruction tuning, it will eventually reach a performance bottleneck. We can use this trend to select some models before reaching the performance bottleneck.

- In multiple rounds of instruction tuning, CITING performed best on the Commonsense Reasoning dataset. Although the performance gain decreases as the instruction tuning rounds increase, the performance gain on the Commonsense Reasoning dataset is the best among all datasets. This shows that on the Commonsense Reasoning dataset, step-by-step reasoning and revision based on criteria and previous responses are very important and can gradually improve the quality of the response.

- We also observe an interesting phenomenon, that is, when there are many rounds of curriculum instruction tuning, the model performance after being fine-tuned deteriorates. Specifically, when curriculum instruction tuning reaches the fourth round, model performance deteriorates in both

| Case 1 | Case 2 |
|---|---|
| **Instruction:** Generate a comment on a science article. | **Instruction:** What skills do you need to be a successful doctor? |
| **CITING Response:** This science article impressively blends meticulous research with clear presentation. The authors skillfully marry empirical data with theoretical insights, catering to both experts and novices. What stands out is the balance between technicality and accessibility. Moreover, the forward-looking discussion not only contextualizes the findings but also hints at future research avenues. Truly, this piece epitomizes rigorous scientific discourse. | **CITING Response:** To be a successful doctor, you must possess key skills. Effective communication and interpersonal abilities are paramount for patient interactions. Quick critical thinking is vital for medical decisions. Being organized and detail-oriented ensures quality patient care. Above all, a genuine passion for aiding others and a continual drive to learn are indispensable. |
| **RLHF Response:** This article was very informative and provided a nice overview of the topic. It provides a detailed view of the current state of the scientific community and the challenges it faces. It also provides some ideas for possible solutions that could be implemented to address the existing problems and ensure the success of the proposed solutions. | **RLHF Response:** A successful doctor requires excellent communication and interpersonal abilities. You must be able to work independently and make decisions quickly. You also must possess strong problem-solving skills and be able to work independently to reach your goals |
| **Ground Truth Response:** This article is a great insight into the latest developments in science. It's amazing to see how far we've come and what scientific discoveries are on the horizon. | **Ground Truth Response:** Success in the medical profession demands a diverse skill set. This encompasses outstanding communication, comprehensive medical knowledge, analytical thinking, and swift decision-making. The ability to handle pressure, manage time effectively, and a genuine empathy for patients are also essential. |
| (a) Case 1. | (b) Case 2. |

Figure 4: Case study for two instructions and the responses from CITING, RLHF and Ground Truth.

the Articulate and Comprehensive metrics of the Reading Comprehension dataset. This is because repeated instruction tuning will cause catastrophic forgetting (Luo et al., 2023; Lin et al., 2023; Korbak et al., 2022) in the model, which makes the fine-tuned model forget the knowledge from the earliest SFT, resulting in a decline in model performance. In addition, we also summarize the revision process of CITING in Appendix C and the criteria summarized by teacher LLM in Appendix B.

## 4.6 CASE STUDY

To analyze the performance of CITING more intuitively, we conduct an empirical case study shown in Figure 4. We analyze cases to verify the rationality of CITING's responses compared with RLHF and Ground Truth, which can direct further improvements.

**Case #1.** From Figure 4(a), we can observe that although the response from RLHF seems specific at a glance, its content is somewhat hollow. This might be because human feedback does not adequately cover instructions of this type. On the other hand, the response from Ground Truth, though concise, is clear in thought. Compared to these two responses, the response from CITING adopts a "whole-part-whole" structure, which is organized and very detailed. This indicates that the revision curriculum helps improve the depth and specificity of the response. This observation validates the experimental results in Section 4.4.

**Case #2.** From Figure 4(b), it can be observed that the sentence patterns of CITING's responses are richer and more diverse, and the overall structure of the paragraph is more complete compared with the responses of RLHF and Ground Truth. This shows that curriculum instruction tuning increases the diversity and flexibility of responses.

## 5 CONCLUSION

In conclusion, this paper presents a novel approach for advancing large language models (LLMs) by addressing the bottleneck of manually crafted instruction datasets and human alignment. We introduce Curriculum Instruction Tuning (CITING), which leverages AI models as teachers to train student LLMs, drawing inspiration from how human students refine their writing skills. This approach involves two key steps: the teacher LLM creates rubrics for evaluating answers to different types of questions, and the student LLM learns to follow these rubrics and self-correct using the teacher's revisions. Our experiments, comparing CITING to state-of-the-art baselines on four datasets, show substantial improvements in terms of articulation, depth, and comprehensiveness.

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

## A  NOTATIONS

We summarize the commonly used notations of our paper in Table 3.

Table 3: A list of commonly used notations.

| Notation | Description |
|---|---|
| $x$ | The instruction. |
| $y$ | The ground truth response. |
| $c$ | The criterion of the instruction. |
| $D$ | The distribution of the data. |
| $x_h$ | The instruction that already has criteria. |
| $x_n$ | The new or test instruction. |
| $E_j$ | The BERT embedding set for each category $j$ of instructions. |
| $\text{Score}_j$ | The similarity score. |
| $f_{\text{SFT}}$ | LLM fine-tuned by SFT. |
| $f_{\text{CITING}}^{(k)}$ | The LLM that is fine-tuned by CITING in the $k-th$ round. |
| $r^{(k)}$ | Response in the $k-th$ round. |
| $pr$ | Prompt designed for curriculum fine-tuning. |
| $\pi^{(k)}$ | The student LLM of $k-th$ round. |
| $N$ | The number of curriculum instruction tuning. |
| $M$ | The number of rounds of inference. |

## B  CRITERIA SUMMARIZED BY TEACHER LLM

We utilize GPT-4 to summarize and classify the criteria in the instructions of the Alpaca dataset, and the results are shown in Table 4. From this table we can observe that the criteria corresponding to instructions are divided into five major categories.

| Category | Criteria |
|---|---|
| Factual Knowledge Instructions | These include questions that require factual responses and can generally be answered with specific, accepted information. A good answer for these types of questions will be accurate, concise, and to the point. |
| Explanation/Definition Instructions | These include questions that require detailed explanations or definitions. Good answers to these questions are typically thorough, logically structured, and free of complex jargon. They make use of clear and understandable language to break down complex topics. |
| Analysis/Evaluation Instructions | These include questions that require some form of analysis or evaluation. A good response will typically be well-reasoned, provide insight, make comparisons where necessary, and may also involve critical thinking. |
| Creative Generation Instructions | These include questions that require creative thinking, such as generating a list, writing a story or poem, or coming up with ideas. Good responses will be original, thoughtful, and fit the given parameters or criteria. |
| Practical Application Instructions | These include questions that require a specific action or task to be performed, such as computation, translation, or conversion. Good responses will accurately complete the task and provide clear, step-by-step reasoning where appropriate. |

Table 4: Criteria summarized by teacher LLM

# C CASES OF INFERENCE PROCEED OF CITING

| Method | Response |
|---|---|
| Instruction | List down five potential names for a new environmentally-friendly energy drink. |
| Initial Response | 1.GreenBurst 2.PureVigor 3.EcoFuel 4.NatureZip 5.EarthBoost |
| $1-th$ Round Revision Prompt | Below is an instruction and its response. In addition, a criteria for the instruction is given to provide a good or bad judgment standard for completing this instruction. Please revise the response according to the given instruction and criteria. ## Instruction: List down five potential names for a new environmentally-friendly energy drink. ## Response: 1.GreenBurst 2.PureVigor 3.EcoFuel 4.NatureZip 5.EarthBoost ## Criteria: These include questions that require creative thinking, such as generating a list, writing a story or poem, or coming up with ideas. Good responses will be original, thoughtful, and fit the given parameters or criteria. ## The revised response is: |
| $1-th$ Round Revision | 1. GreenBurst - Derived from the lush colors of the forest. PureVigor - Emphasizes the natural purity and energy it provides. 2. BioBlast - Highlights its eco-friendly and powerful characteristics. 3. NatureZip - Fast-paced, just like nature's rapid processes. 4.TerraTonic - A play on Earth (Terra) and the revitalizing properties of a tonic. |
| $2-th$ Round Revision Prompt | Below is an instruction and its response. In addition, a criteria for the instruction is given to provide a good or bad judgment standard for completing this instruction. Please revise the response according to the given instruction and criteria. ## Instruction: List down five potential names for a new environmentally-friendly energy drink. ## Response: 1. GreenBurst - Derived from the lush colors of the forest. PureVigor - Emphasizes the natural purity and energy it provides. 2. BioBlast - Highlights its eco-friendly and powerful characteristics. 3. NatureZip - Fast-paced, just like nature's rapid processes. 4.TerraTonic - A play on Earth (Terra) and the revitalizing properties of a tonic. ## Criteria: These include questions that require creative thinking, such as generating a list, writing a story or poem, or coming up with ideas. Good responses will be original, thoughtful, and fit the given parameters or criteria. ## The revised response is: |
| $2-th$ Round Revision | 1. GreenBurst Radiance - Inspired by the green forest, it radiates energy. 2. PureVigor Essence - Harnesses the essence of nature for pure energy. 3. BioBlast Surge - A surge of energy with a nod to its biodegradable nature. 4. NatureZip Pulse - Capturing nature's heartbeat and energy in every sip. 5. TerraTonic Revive - Refreshing the spirit with Earth's finest ingredients. |
| Instruction | Write a short poem inspired by the tranquility of a quiet evening. |
| Initial Response | The evening comes, and the world grows still, The setting sun rests behind the hill, Stars twinkle, a silent lullaby, As day turns to night, with a soft sigh. |
| $1-th$ Round Revision Prompt | Below is an instruction and its response. In addition, a criteria for the instruction is given to provide a good or bad judgment standard for completing this instruction. Please revise the response according to the given instruction and criteria. ## Instruction: Write a short poem inspired by the tranquility of a quiet evening. ## Response: The evening comes, and the world grows still, The setting sun rests behind the hill, Stars twinkle, a silent lullaby, As day turns to night, with a soft sigh. ## Criteria: These include questions that require creative thinking, such as generating a list, writing a story or poem, or coming up with ideas. Good responses will be original, thoughtful, and fit the given parameters or criteria. ## The revised response is: |
| $1-th$ Round Revision | The evening drapes, as shadows spill, Golden hues retreat, behind the tranquil hill, Stars twinkle, composing night's lullaby, Whispered tales of day's end, a muted sky-high. |
| $2-th$ Round Revision Prompt | Below is an instruction and its response. In addition, a criteria for the instruction is given to provide a good or bad judgment standard for completing this instruction. Please revise the response according to the given instruction and criteria. ## Instruction: Write a short poem inspired by the tranquility of a quiet evening. ## Response: The evening drapes, as shadows spill, Golden hues retreat, behind the tranquil hill, Stars twinkle, composing night's lullaby, Whispered tales of day's end, a muted sky-high. ## Criteria: These include questions that require creative thinking, such as generating a list, writing a story or poem, or coming up with ideas. Good responses will be original, thoughtful, and fit the given parameters or criteria. ## The revised response is: |
| $2-th$ Round Revision | Silken dusk, as the world takes its fill, Amber retreats, a tranquil hill's window sill, Galaxies dance, in a cosmic lullaby, Bidding adieu to the day, with a serene sky-high. |

Table 5: Comparison of different responses for the same instruction

## D  THE RATIONALITY OF EMBEDDING SIMILARITY-BASED RUBRIC

1.In order to verify the fact that the embedding of candidate instructions is close to the title means that the candidate instructions are legal, we designed experiments for 1000 queries to verify. We first let GPT-4 determine its criteria for these 1000 queries. Then take out 100 of them as data for known criteria, and the remaining 900 as test data for unknown criteria. Our experiments found that the method based on embedding similarity matching can achieve a criterion prediction accuracy of 72% on 900 test data sets, which proves the effectiveness of our method. 2.Although simply using rubrics as "system message" in the few-shot examples can simplify CITING's pipeline, it has two problems: 1. Since the query exists in various forms in the data set, the "system message" needs to be very complex. To achieve a good effect, a large number of examples need to be given, and the limitations of the context window of student LLM make this approach impossible to implement; 2. The corresponding information of query and criteria contained in the example in "system message" also requires student LLM to learn, which increases the difficulty of student LLM learning compared to directly matching the criteria of the corresponding query.

## E  IMPLEMENTATION DETAIL OF RLHF

To ensure a fair comparison of all baselines, we first we first use the original llama-1 model generate two different responses for 1,000 queries from the training dataset. We then use GPT-3.5 to rank the quality of these two responses for each query. Next, we train a reward model based on this data and implement RLHF.

