# OpenReview forum: "CITING: Large Language Models Create Curriculum for Instruction Tuning"
_ICLR.cc/2024/Conference — Submitted to ICLR 2024_

### Official Review · Reviewer_ygGy · 2023-10-31

**Soundness:** 2 fair
**Presentation:** 3 good
**Contribution:** 3 good
**Rating:** 5
**Confidence:** 4

**Summary:**

This paper proposes an alternative to RL-based methods (e.g.: RLHF) for improving the quality of generations from instruction-tuned models by distilling preferences from other language models and using them in a supervised finetuning setup. Particularly, a teacher LLM is used to improve a student LLM by 1) generating a rubric (or criteria for good responses) for each instruction in an instruction-tuning dataset, and 2) iteratively rewriting the student's response given the rubric and the current response.

The student is initialized by training it on the instruction tuning dataset using SFT and then continued to be trained to rewrite its responses given the targets from the teacher. At inference time, a rubric is retrieved from the set generated for the training data, and an iterative process is followed to generate the student's response.

This process is used to train a Llama-7B model on the Alpaca dataset, and it is evaluated against SFT, RLHF,  RRHF (Yuan et al., 2023; RLHF with ranking instead of pairwise preferences) and RAFT (Dong et al. 2023; use reward model to select training data) in terms of win-rate according to GPT-4 on a held-out subset of Alpaca, reading comprehension, factual knowledge, and commonsense datasets.

**Strengths:**

Exploring alternatives to RL algorithms using sparse feedback to improve LM generations is highly relevant and timely. This paper presents some good ideas, particularly using feedback from other LMs in the form of rubrics as rewrites, that can contribute to this research area.

**Weaknesses:**

Algorithm: The student is trained as a response rewriter given a rubric during the iterative process. This raises the following concerns:

- Because of this formulation the student model requires a rubric and a version of the response to rewrite at inference time. The dependence on additional inputs might impact the generalizability of the student model as obtaining. Particularly, the rubrics generated for the training data may not generalize to the new instructions at inference time.
- Moreover, the inference process is required to be iterative due to this algorithm, and hence requires additional compute.
- Since the student is initialized using SFT on the instruction tuning dataset, the initial draft responses produced by the model may be of good quality, but continuing to train it to be a rewriter might affect the original instruction following capabilities. An evaluation of the original drafts produced by the student after the iterative training might be helpful.

One solution to address the concerns above could be to train a separate rewriter (conditioned on the rubric and the previous responses), and iteratively finetune the student model only on the instruction following task (i.e., not condition its outputs on the rubric and previous responses)

Baselines: This paper proposes multiple changes compared to the current RL from feedback setup for improving instruction tuning models: having the teacher generate a rubric, continued training of models with SFT, curriculum learning with minimal rewrites given by a teacher. These are all orthogonal improvements and can be evaluated separately. The current evaluation setup conflates these changes. I propose the following additional baselines:
- Effect of continued training: Simply train the student model with additional SFT steps, and do not use a teacher model at all. The current comparison between CITING and SFT is not entirely fair because CITING uses additional training steps.
- Effect of the rubrics: Use the teacher model to generate a rubric and train the student model with additional SFT steps also conditioned on the rubric.
- Fairer comparison with RLHF: RLHF might perform better with better pairwise preference data. A good baseline would be to generate rewrites from a teacher model and use them to train a reward model and perform RLHF.

**Questions:**

- The details of the baselines are missing. How exactly are the RLHF models trained? Do they use preferences on the same 1000 instances used for CITING?
- How do you ensure that the teacher minimally edits the student responses in the CITING algorithm? Do you actually see that the edits are minimal?
- How do you select the number of iterations at training time and inference time in CITING?

---

> ### Author Response · Authors · 2023-11-22
> **Response to the Reviewer ygGy (1/2)**
>
> We sincerely thank reviewer **ygGy**’s valuable time and comments. We provide point-wise responses below.
> **Q1.** The details of the baselines are missing. How exactly are the RLHF models trained? Do they use preferences on the same 1000 instances used for CITING?
>
> **Response:**  Thanks for noticing the missing implementation details in the paper. Specifically, to ensure a fair comparison of all baselines, we first use the original llama-1 model to generate two different responses for 1,000 queries from the training dataset. We then use GPT-3.5 to rank the quality of these two responses for each query. We then utilize these data to  implement RLHF. We will add these explanations in Appendix G.
>
>
> **Q2.** How do you ensure that the teacher minimally edits the student responses in the CITING algorithm? Do you actually see that the edits are minimal?
>
> **Response:**  This is a good point. Our method does not guarantee that teachers can minimally edit the student responses. In fact, the CITING algorithm is to build a curricularly revision mechanism so that student LLM can learn how to modify its initial answer step by step with the help of teacher LLM. Of course, such a mechanism does not guarantee that every teacher's modification to the student's response will be minimal. This modification may be large or small. We have tried to add an additional prompt to the teacher model that "it should make minimal edits to the student model" and it can work for some cases. But when the student model's outputs are totally wrong are meaningless, the edits can still be equal to rewriting.
>
>
> **Q3.** How do you select the number of iterations at training time and inference time in CITING?
>
> Response:We set up a validation set during both the training and testing stages to help us choose the number of iterations. We select the number of iterations with the best performance on the validation set as the final result.
>
>
> **Q4.** Algorithm: The student is trained as a response rewriter given a rubric during the iterative process. This raises the following concerns:
> * Because of this formulation the student model requires a rubric and a version of the response to rewrite at inference time. The dependence on additional inputs might impact the generalizability of the student model as obtaining. Particularly, the rubrics generated for the training data may not generalize to the new instructions at inference time.
> * Moreover, the inference process is required to be iterative due to this algorithm, and hence requires additional compute.
> * Since the student is initialized using SFT on the instruction tuning dataset, the initial draft responses produced by the model may be of good quality, but continuing to train it to be a rewriter might affect the original instruction following capabilities. An evaluation of the original drafts produced by the student after the iterative training might be helpful.
> One solution to address the concerns above could be to train a separate rewriter (conditioned on the rubric and the previous responses), and iteratively finetune the student model only on the instruction following task (i.e., not condition its outputs on the rubric and previous responses)
>
> **Response:**  Since the queries in the Alpaca data set are diverse and cover a wide range of types, we consider selecting a certain amount of queries so that the criteria obtained by GPT-4 can have better generalization. In addition, we agree with what you said about continuing to train the student LLM to be a rewriter might affect the original instruction following capabilities. However, our experimental results show that this method still shows good performance. To train a separate rewriter  is indeed a good way to solve the above problems, but this method requires a lot of cost of  teacher LLM, and training a separate rewriter independently also requires additional network cost. Thank you very much for your suggestions. We will compare your proposed solutions in the final version and comprehensively analyze the advantages and disadvantages of our methods.

---

> > ### Author Response · Authors · 2023-11-22
> > **Response to the Reviewer ygGy (2/2)**
> >
> > **Q5.** Baselines: This paper proposes multiple changes compared to the current RL from feedback setup for improving instruction tuning models: having the teacher generate a rubric, continued training of models with SFT, curriculum learning with minimal rewrites given by a teacher. These are all orthogonal improvements and can be evaluated separately. The current evaluation setup conflates these changes. I propose the following additional baselines:
> >
> > * Effect of continued training: Simply train the student model with additional SFT steps, and do not use a teacher model at all. The current comparison between CITING and SFT is not entirely fair because CITING uses additional training steps.
> > * Effect of the rubrics: Use the teacher model to generate a rubric and train the student model with additional SFT steps also conditioned on the rubric.
> > * Fairer comparison with RLHF: RLHF might perform better with better pairwise preference data. A good baseline would be to generate rewrites from a teacher model and use them to train a reward model and perform RLHF.
> >
> > **Response:**  Thank you very much for your suggestions. We will include comparisons with these baselines in the final version.

---

### Official Review · Reviewer_TqZ2 · 2023-11-01

**Soundness:** 3 good
**Presentation:** 3 good
**Contribution:** 3 good
**Rating:** 6
**Confidence:** 3

**Summary:**

This paper proposes a method to train large language models using AI feedback instead of human feedback. The method, called Curriculum Instruction TunING (CITING), uses a teacher LLM to create rubrics and revisions for different types of instructions, and a student LLM to learn from them. The paper shows that CITING outperforms existing methods on four datasets in terms of articulation, depth, and comprehensiveness.

**Strengths:**

1. This paper proposes a novel method to train large language models using AI feedback instead of human feedback, which reduces the cost and difficulty of scaling LLM development.
2. This paper introduces curriculum instruction tuning, which leverages a teacher LLM to create rubrics and revisions for different types of instructions, and a student LLM to learn from them. This is an interesting use case of LLM as a planner for training another LM.
3. This shows that CITING outperforms existing methods on four datasets in terms of articulation, depth, and comprehensiveness.

**Weaknesses:**

1. The technical novelty may be limited. The method is also complicated.
2. This paper does not evaluate the robustness or generalization of CITING to unseen or adversarial instructions. This could be an issue as the teacher model only teaches in-domain curriculums. I'd like to see discussion on this.

**Questions:**

See weaknesses.

---

> ### Author Response · Authors · 2023-11-22
> **Response to the Reviewer TqZ2**
>
> We sincerely thank reviewer **TqZ2**’s valuable time and comments. We provide point-wise responses below.
>
> **Q1.** The technical novelty may be limited. The method is also complicated.
>
> **Response:**
> We thanks for the reviewer to propose the limitations of this work. Though we are not clear which parts the reivewer refers to, we hope to clarify more on the novelty and complexity of the proposed method below.
>
> * Novelty: Compared with the previous learning-from-AI-feedback (LAIF) method, CITING can more efficiently utilize the information fed back by teacher LLM. The traditional LAIF method often only provides the query response to the student LLM. On this basis, CITING allows student LLM to provide the response to the query and also provides the criteria to answer the query. Therefore, CITING not only teaches student LLM the answer to the question, but also teaches it how to answer and correct the question than the traditional LAIF. Compared with the previous synthetic instruction tuning method (i.e., ask the teacher LLM to generate synthetic instruction data), CITING makes full use of human written answers and prompts the teacher LLM to supervise the student model to improve their outputs. This alignment process is free of reinforcement learning as used in RLHF thus being easy to implement and more robust.
>
> * Complexity：We add little complexity in CITING compared with the classic instruction tuning process. There is one more step that we need to call a teacher model to provide feedback to the student model to improve the outputs of the student. Compared to other alignment approaches, such as RLHF, CITING does not involve an additional reward model and reinforcement learning steps, thus being much easier to implement and computationally friendly.
>
>
> **Q2.** This paper does not evaluate the robustness or generalization of CITING to unseen or adversarial instructions. This could be an issue as the teacher model only teaches in-domain curriculums. I'd like to see discussion on this.
>
> **Response:**  We agree that improving the robustness and safety of the student LLM is crucial to deploying LLM in production. Since human alignment is expensive and time-consuming, this paper proposes to use a powerful teacher LLM to supervise the student model, referring to the human written answers. In our experiments, the training set is the Alpaca dataset. We then tested not only on the test set of the equally distributed Alpaca dataset but also on unseen instruction datasets such as World Knowledge, Reading Comprehension, and Commonsense Reasoning. Experimental results show that our method performs much better than other baselines on identically distributed data sets and unseen or adversarial instructions data sets, proving the generalization of CITING.
>
>
> On the other hand, since it is hard to make a student model as powerful as the teacher model in all aspects due to the parameter size limits, CITING can be used to build a specialized student model that is enhanced with the given instruction dataset and the supervision from the teacher model. In this sense, we can also apply CITING to specific sets of curriculums to develop such specialized student model.

---

### Official Review · Reviewer_4C6N · 2023-11-01

**Soundness:** 2 fair
**Presentation:** 2 fair
**Contribution:** 2 fair
**Rating:** 3
**Confidence:** 5

**Summary:**

The paper proposes an instruction tuning method, which first employs a teacher LLM to craft the rubrics for evaluating the answers to various types of questions, and then trains the student LLM to follow the rubrics and perform self-correction from the revision by the teacher. The authors have shown its effectiveness on four datasets, comparing with several baseline methods.

**Strengths:**

**Clarity**

- The presentation of this work is clear and easy to follow.
- The method is simple and effective and has shown clear improvement over baselines.

**Weaknesses:**

My main concern about this work is about novelty and clarity. Even though the method proposes using criteria as a guidance to augment instruction tuning data, the overall method, still, can be viewed as a complex version of data distillation. Recently there have been tons of works proposing pretty similar ideas, such as Orca [1], WizardLM [2], MAmmoTH [3], etc., which all leverage data augmentation (guided by certain criteria/score function, etc.). It might be good to compare CITING with these methods, or at least discuss why the contribution of this work is significant.

I'm not fully convinced about the method of how you assign quality scores to the generated data. In Section 3.2, why does the fact that the candidate instructions have their embeddings near those of rubrics mean that the candidates are legit? In terms of the Appendix, the so-called rubrics are basically some descriptions or requirements of a certain task --- do we really need a complex pipeline like CITING or we can simply include them as the "system message" in the few-shot examples (for augmentation)?

Also, there is no code and data uploaded as supplementary materials, which causes some difficulties for me to fully understand your method.

[1] Orca: Progressive Learning from Complex Explanation Traces of GPT-4

[2] WizardLM: Empowering Large Language Models to Follow Complex Instructions

[3] MAmmoTH: Building Math Generalist Models through Hybrid Instruction Tuning

**Questions:**

How did you implement RLHF? There is no official implementation of that and your method has shown significant improvement over RLHF. Could you upload your code if possible (or point to some open-sourced implementation you are using)?

---

> ### Author Response · Authors · 2023-11-22
> **Response to the Reviewer 4C6N (1/2)**
>
> We sincerely thank reviewer **4C6N**’s valuable time and comments. We provide point-wise responses below.
>
> **Q1.**  My main concern about this work is about novelty and clarity. Even though the method proposes using criteria as a guidance to augment instruction tuning data, the overall method, still, can be viewed as a complex version of data distillation. Recently there have been tons of works proposing pretty similar ideas, such as Orca [1], WizardLM [2], MAmmoTH [3], etc., which all leverage data augmentation (guided by certain criteria/score function, etc.). It might be good to compare CITING with these methods, or at least discuss why the contribution of this work is significant.
> [1] Orca: Progressive Learning from Complex Explanation Traces of GPT-4
> [2] WizardLM: Empowering Large Language Models to Follow Complex Instructions
> [3] MAmmoTH: Building Math Generalist Models through Hybrid Instruction Tuning
>
> **Response:**  Thanks for the reviewer to propose those relevant papers from the literature! However, after taking a deep look into these papers, we believe CITING is fundamentally different from them.
>
> First, these methods essentially try to distill the capability from the teacher model by training the student model on the synthetic instruction data from the teacher. In this process, no human written answers are involved, and there is limited control of hallucinations and errors in the synthetic instruction datasets. By contrast, CITING leverages the teacher model to generate the revisions from the student model referring to the human written answers, alleviating the hallucination issues. Also, the student model in CITING is a distilled teacher model and specialized in a specific scope of tasks, thus being possible to be better than the teacher while the other synthetic instruction tuning method cannot yield such a student model but causes the model to deteriorate (mode collapse) over time, which has been proposed by recent papers **[R1]**.
>
> Second, there is no alignment for the student model in the other synthetic instruction tuning method but CITING focuses on how to optimize the student LLM step by step with the feedback from teacher LLM. During this process, the teacher LLM will receive the last revision of the student LLM and conduct revisions and guides based on the criteria, thus establishing communication and feedback between the two.
>
> **[R1]** Shumailov I, Shumaylov Z, Zhao Y, et al. The Curse of Recursion: Training on Generated Data Makes Models Forget[J]. arXiv preprint arxiv:2305.17493, 2023.
>
>
> **Q2.**  I'm not fully convinced about the method of how you assign quality scores to the generated data. In Section 3.2, why does the fact that the candidate instructions have their embeddings near those of rubrics mean that the candidates are legit? In terms of the Appendix, the so-called rubrics are basically some descriptions or requirements of a certain task --- do we really need a complex pipeline like CITING or we can simply include them as the "system message" in the few-shot examples (for augmentation)?
>
> **Response:**  Thanks for pointing out the parts that cause confusion. We hope to add the following clarifications.
>
> 1) In order to verify the fact that the embedding of candidate instructions is close to the title means that the candidate instructions are legal, we designed experiments for 1000 queries to verify. We first let GPT-4 determine its criteria for these 1000 queries. Then take out 100 of them as data for known criteria, and the remaining 900 as test data for unknown criteria. Our experiments found that the method based on embedding similarity matching can achieve a criterion prediction accuracy of 72% on 900 test data sets, which proves the effectiveness of our method.
>
> 2) Although simply using rubrics as "system message" in the few-shot examples can simplify CITING's pipeline, it has two problems: 1. Since the query exists in various forms in the data set, the "system message" needs to be very complex. To achieve a good effect, a large number of examples need to be given, and the limitations of the context window of student LLM make this approach impossible to implement; 2. The corresponding information of query and criteria contained in the example in "system message" also requires student LLM to learn, which increases the difficulty of student LLM learning compared to directly matching the criteria of the corresponding query. We have added this part of the discussion to Appendix D of the paper.

---

> > ### Author Response · Authors · 2023-11-22
> > **Response to the Reviewer 4C6N (2/2)**
> >
> > **Q3.**  How did you implement RLHF? There is no official implementation of that and your method has shown significant improvement over RLHF. Could you upload your code if possible (or point to some open-sourced implementation you are using)?
> >
> > **Response:**  We have upload the  implementation of our RLHF to the supplementary material. Specifically, to ensure a fair comparison of all baselines, we first we first use the original llama-1 model generate two different responses for 1,000 queries from the training dataset. We then use GPT-3.5 to rank the quality of these two responses for each query. Next, we train a reward model based on this data and implement RLHF. We have added this part of the discussion to Appendix E of the paper.

---

> > > ### Comment · Reviewer_4C6N · 2023-11-23
> > > **Thanks for your response. Follow-up questions.**
> > >
> > > Thanks for your response. I still feel confused about why embedding-based method could work. Could you upload the so-called 1000 samples (100 for known criteria + 900 unknown criteria) which seem to serve as a pilot data pool for the model to decide the criteria, and the code of your method? I might want to try the method locally to see whether we can use the embedding of rubrics and the candidate data samples to filter qualified data.
> > >
> > > Some follow-up questions are:
> > > 1. if you can already find those data that meet your criteria, why not just use self-instruct to augment more data (then the embedding calculation should be between the candidate data and the verified data)?
> > > 2. Are there failure cases in CITING, or error analysis?
> > >
> > > Thanks!

---

> > > > ### Author Response · Authors · 2023-11-23
> > > > **Response to the Reviewer 4C6N**
> > > >
> > > > Dear reviewer:
> > > >
> > > > **1.** We appreciate your interest in our work. To provide further clarity, we have included 1000 samples along with the corresponding criteria similarity matching code in the supplementary material. We kindly invite you to review these for more detailed information.
> > > >
> > > > **2.** Thank you for your query regarding our methodology. To elucidate, we employ a teacher LLM to impart criteria to the student LLM, thereby enhancing the latter's ability to incrementally refine its initial responses based on these criteria. This methodology is detailed in Figure 3:
> > > >
> > > > Initially, the teacher LLM categorizes the query using examples from the instruction data and establishes criteria for the response quality in each category. Subsequently, for new queries with unknown criteria, we apply similarity matching to align the query with the appropriate criteria. This approach enables us to direct the student LLM in revising responses, both in the training and testing phases, for queries lacking predefined criteria. It's important to note that our process does not involve self-instruction or the use of verified data.
> > > >
> > > > **3.**  We acknowledge the presence of failure cases in our CITING model. These primarily occur when the model's initial response to a query contains significant errors or produces outputs that lack meaningful content, which in turn limits the effectiveness of subsequent revisions. However, it's important to note that such instances constitute less than 1% of our overall results. Furthermore, during the training phase of CITING, even when the student LLM's outputs are incorrect or nonsensical, the teacher LLM's edits can effectively amount to a complete rewrite. This mechanism plays a crucial role in mitigating the frequency of failure cases to a certain degree.

---

### Official Review · Reviewer_6BWB · 2023-11-02

**Soundness:** 3 good
**Presentation:** 3 good
**Contribution:** 3 good
**Rating:** 6
**Confidence:** 4

**Summary:**

The paper introduces a novel approach named Curriculum Instruction TunING (CITING) for the development and scaling of Large Language Models (LLMs). Instead of a heavy reliance on human-crafted instruction datasets and human alignment, CITING employs a teacher LLM to guide and train student LLMs. This methodology mirrors the traditional tutor-student dynamic, where students refine their skills using rubrics and revisions. The process entails the teacher LLM offering evaluation criteria, with the student LLM subsequently learning to self-correct based on these guidelines. Experimental findings indicate that CITING surpasses contemporary leading methods like RLHF across several benchmarks.

**Strengths:**

1. The Curriculum Instruction TunING approach is innovative. Using teacher LLMs to guide student LLMs, which mirrors the tutor-student relationship, is a fresh perspective in this field.

2. The paper delineates a meticulously crafted methodology, ranging from rubric design with the teacher model to the iterative fine-tuning of the student LLM.

3. The narrative is lucid, providing a thorough explanation of the CITING methodology.

**Weaknesses:**

1. Over-reliance on Teacher LLM: There's a potential risk if the teacher LLM possesses biases or inaccuracies, as it could transfer these shortcomings to the student LLM. Consequently, the effectiveness of CITING is largely contingent on the quality and resilience of the teacher LLM.

2. Test Phase Limitations: During the test phase, the model's potential might be constrained by the extent of criteria it can retrieve from a fixed corpus.

3. Evaluation Metrics: The paper predominantly emphasizes the winning rate for comparing with other techniques. Yet, in scenarios like QA or RC, shouldn't the method also be evaluated using standard metrics?

**Questions:**

1. How does CITING address potential biases or inaccuracies if the teacher LLM possesses them?

2. During the test phase, how does the model overcome the limitations of retrieving criteria from a fixed corpus?

3. Beyond the winning rate, have other standard metrics been considered for evaluating CITING, especially in QA or RC scenarios?

---

> ### Author Response · Authors · 2023-11-22
> **Response to the Reviewer 6BWB**
>
> We sincerely thank reviewer **6BWB**’s valuable time and comments. We provide point-wise responses below.
>
> **Q1.** How does CITING address potential biases or inaccuracies if the teacher LLM possesses them?
>
> **Response:** Thanks for pointing out this concern. The current CITING does not have a dedicated module to consider potential biases or inaccuracies if the teacher LLM possesses. We will consider adding some feedback mechanisms based on CITING to solve this problem. Nonetheless, we conjecture the bias and inaccuracies should be minor since we are not asking the teacher LLM to make the revision from scratch. Instead, the teacher LLM's input also includes the groundtruth answers, so it is prompted to improve the student's input referring to the groundtruth.
>
> **Q2.**  During the test phase, how does the model overcome the limitations of retrieving criteria from a fixed corpus?
>
> **Response:**  When we generated fixed criteria, we asked GPT-4 to summarize the problems in the training set and summarized them based on the characteristics of these problems to obtain criteria for different categories of problems. Therefore, we believe that such criteria are representative and comprehensive and should be able to cover all the problems in the test set. From a cost perspective, considering a fixed criteria corpus can avoid the need to generate specific criteria for different problems on the fly. More importantly, CITING is proposed to distill and enhance the capability of the teacher model in specialized application scenarios. The fine-tuned student mode is expected to handle a fixed and specialized scope. In this sense, there is no need to adapt the criteria.
>
>
>
> **Q3.** Beyond the winning rate, have other standard metrics been considered for evaluating CITING, especially in QA or RC scenarios?
>
> **Response:**  We thanks the reviewer to propose more evaluation metrics. There are indeed many NLP standard metrics that can be used to evaluate CITING, such as PPL, BLEU and Precision. However, these standard metrics are evaluated by measuring the similarity between the response generated by LLM and the ground truth. However, they are very limited in actual QA and RC tasks, because LLM may generate responses that are equally reasonable but very different from the ground truth. There are now many recent works that introduce GPT-4 to conduct a more comprehensive evaluation of the responses generated by LLM from various perspectives **[R1,R2]**. Therefore, in this paper, we follow this setting and utilize GPT-4 as our evaluation metric.
>
> **[R1]** Dong, Hanze, et al. "Raft: Reward ranked finetuning for generative foundation model alignment." arXiv preprint arXiv:2304.06767 (2023).
>
> **[R2]** RRHF: Rank Responses to Align Language Models with Human Feedback without tears

---

### Meta-Review · Area_Chair_LFQx · 2023-12-06

**Metareview:**

The paper presents an innovative approach to instruction tuning of Large Language Models (LLMs) by employing a teacher LLM to guide and train student LLMs. The reviewers appreciated the novelty of the approach and the clarity of the presentation. However, they raised concerns about the over-reliance on the teacher LLM, the limitations during the test phase, and the evaluation metrics used.

The reviewers also questioned the novelty of the method, suggesting that it could be viewed as a complex version of data distillation, and requested comparisons with similar methods. There were also concerns about the assignment of quality scores to the generated data and the lack of supplementary materials, such as code and data.

The paper's exploration of alternatives to RL algorithms using sparse feedback to improve LM generations was seen as a valuable exploration. However, the reviewers raised concerns about the algorithm, the dependence on additional inputs, the iterative inference process, and the impact on the original instruction following capabilities. They also suggested additional baselines for a fairer comparison.

Given these concerns and suggestions, the recommendation is to reject the paper.

**Justification For Why Not Higher Score:**

There are several concerns that prevent a higher score.

1. The over-reliance on the teacher LLM is a significant concern. If the teacher LLM has biases or inaccuracies, these could be transferred to the student LLM, potentially affecting the effectiveness of the proposed method.

2. The evaluation metrics used in the paper were questioned by the reviewers. The paper primarily uses the winning rate for comparison with other techniques, but the reviewers suggested that standard metrics should also be used in scenarios like QA or RC.

3, the novelty of the method was questioned, with some reviewers suggesting that it could be viewed as a complex version of data distillation. They requested comparisons with similar methods to better understand the contribution of this work.

**Justification For Why Not Lower Score:**

N/A

---

### Decision · Program_Chairs · 2024-01-16

Reject